# The Effect of the Thermal Annealing Process to the Sensing Performance of Magnetoelastic Ribbon Materials

**Georgios Samourgkanidis** **, Kostantis Varvatsoulis and Dimitris Kouzoudis ***

Department of Chemical Engineering, University of Patras, GR 26504 Patras, Greece;
G.Samourgkanidis@gmail.com (G.S.); kostantis.varvatsoulis@gmail.com (K.V.)
* Correspondence: kouzoudi@upatras.gr

**Abstract:** The magnetoelastic materials find many practical applications in everyday life like transformer cores, anti-theft tags, and sensors. The sensors should be very sensitive so as to be able to detect minute quantities of miscellaneous environmental parameters, which are very critical for sustainability such as pollution, air quality, corrosion, etc. Concerning the sensing sensitivity, the magnetoelastic material can be improved, even after its production, by either thermal annealing, as this method relaxes the internal stresses caused during manufacturing, or by applying an external DC magnetic bias field during the sensing operation. In the current work, we performed a systematic study on the optimum thermal annealing parameters of magnetoelastic materials and the Metglas alloy 2826 MB3 in particular. The study showed that a 100% signal enhancement can be achieved, without the presence of the bias field, just by annealing between 350 and 450 °C for at least half an hour. A smaller signal enhancement of 15% can be achieved with a bias field but only at much lower temperatures of 450 °C for a shorter time of 20 min. The magnetic hysteresis measurements show that during the annealing process, the material reorganizes itself, changing both its anisotropy energy and magnetostatic energy but in such a way such that the total material energy is approximately conserved.

**Keywords:** magnetoelastic sensors; Metglas; annealing process; hysteresis loop; optimization process

## 1. Introduction

Sensors are an integrable part of a modern society as they help to continuously monitor vital physical/chemical parameters, which are the crucial components for society's long-range sustainability. For example, sensors help with smart urbanization [1], monitor the air quality [2] and the water quality [3], help against pollution by monitoring sewer networks [4], and improve production in agriculture [5,6].

On the other hand, magnetoelastic materials are advanced materials that are used in smart sensor technology due to their unique properties and low cost. For example, they are used heavily as anti-theft tags on commercial products such as clothing and books (the white little tag that is attached on the internal side of the back cover of the book). Their unique properties stem from the fact that these materials deform elastically under the presence of an external magnetic field, and vice-versa; when they are subject to mechanical straining, they develop magnetization. These materials are usually made of amorphous metallic alloys, widely known as metallic glasses or Metglas for short [7]. Due to both their soft magnetic properties, which practically mean a linear relationship between the applied magnetic field and the resulting magnetization, and their amorphous nature, which means high resistivity and low eddy currents, they are excellent candidates for transformer applications, giving a linear input–output relationship with low losses (see transformer products by Metglas Inc. located in Conway, SC, USA). As sensing devices, they find great usage in a variety of applications such as vibration sensors [8–12] or as mass-load sensors [13,14] for bio-sensing [15–19] or gas-sensing [20–22] applications. Furthermore,

these materials are the magnetic analogs of electric piezo-crystals, which, for example, are utilized to detect very weak magnetic fluctuations and where the magnetic response optimization is equally critical [23,24].

Of crucial importance in these applications is the optimization of the material properties of Metglas so as to produce the maximum response. Typically, there are two basic improvements that can be done towards this direction. In applications where the material is set to vibrate by either external alternating strains or external alternating magnetic fields, the additional application of a DC bias field reduces the internal anisotropy field, leading to an easier alignment of the magnetic domains and thus to a higher magnetization. The other improvement is the thermal annealing of these materials, prior to their use, which eliminates the internal stresses that are typically developed during their synthesis. This process also reduces the internal anisotropy field [25,26]. Part of the latter case is also the way to increase the material performance by annealing the sensor while exposed to a transverse magnetic field [27–30].

Some annealing studies have been done in the past for different Metglas alloys. More specifically, Benda and Bydžovský [31] studied Metglas 2605-S2, with an average composition of $Fe_{78}B_{13}Si_9$ and a Curie temperature of $T_c = 415$ °C. They investigated the effect of annealing by comparing three different samples, a ribbon as cast, one annealed for 25 min at 300 °C, and one annealed for 1 h at 345 °C. The results revealed an increase in the saturation magnetization $M_s$ by a factor of 2 and a decrease in the coercive force $H_c$ by a factor of 5 at 345 °C and proportionally smaller changes at 300 °C. Celasco et al. [32] worked on Metglas 2826, with an average composition $Fe_{40}Ni_{40}B_{14}B_6$, and measured the power losses in the frequency range of 2–20 kHz. They found that thermal annealing has a very large effect on the dynamic power loss. The loss reduction is strongly enhanced by increasing the annealing temperature up to 350 °C. Higher annealing temperatures begin to produce crystallization effects and increase the static loss. Gutiérrez et al. [33] worked on $Fe_{64}Ni_{10}Nb_3CuSi_{13}B_9$ samples, which they annealed in a vacuum for 1 h in the temperature range of 300–550 °C. They found that the coercivity $H_c$ decreases by a factor of 10 upon annealing, and the effective anisotropy K decreases by a factor of 5, going from the as-quenched state to the annealed sample at 520 °C. Huang et al. [34] worked on Metglas 2826 MB, with an average composition $Fe_{45}Ni_{45}Mo_7B_3$ and $T_c = 353$ °C. Annealing was carried out at 70, 150, 200, 250, and 300 °C for 2 h under a vacuum. They found a slight increase in the quality factor Q at about 10% at the optimum annealing temperature of 250 °C. At this temperature, SEM micrographs reveal that all microcracks disappear indicating a stress relief situation. Yang et al. [35] worked on Metglas Alloy 2605SA1 ($Fe_{78}Si_{13}B_9$) with $T_c = 508$ °C. They tried 150 °C to 450 °C in steps of 100 °C, and, contrary to other authors, they did not see any improvement in the magnetization characteristics up to the critical temperature, while they saw degradation (about 30% lower saturation magnetization) as expected, by performing annealing at 550 °C, which is well above the Curie temperature. Sahingoz et al. [36] worked on Metglas 2605-S2, with an average composition of $Fe_{78}B_{13}Si_9$ and $T_c = 415$ °C. They annealed at 500 °C for different time intervals. Unfortunately, they only annealed well above $T_c$, which usually leads to a worsening of the soft properties of the material; for example, their coercivity field $H_c$ increased overall by a factor of 11 after 50 min, even though initially there was a slight decrease by a factor of 0.5 at a short interval time of 3 min.

As the thermal annealing of the magnetoelastic materials is a very crucial process for the maximum performance as smart sensing devices, we feel that a thorough study is of paramount importance for the people working in the field, both in industry and in academia. The current work is aiming towards this direction, providing a complete study on thermal annealing for the Metglas alloy 2826 MB3 ($Fe_{40}Ni_{38}Mo_4B_{18}$) with $T_c = 353$ °C, which is currently the most dominant magnetoelastic material for sensor applications.

## 2. Experimental Setup and Procedure

A reel of Metglas alloy 2826 MB3 with width 1/4 inch and thickness 30 μm was purchased by Metglas, Inc. Shown in Figure 1a is the set of ribbons cut out of this reel and used in the current work. Each ribbon was cut in the same length of 2.5 cm and annealed at different temperatures and different times, as shown by the numbers in the figure. The annealing process was as follows: The oven was given enough time to reach a fixed temperature, for example, 50 °C, and then a group of six ribbons were inserted into it, with each ribbon corresponding to a different annealing time between 10 to 60 min with a step of 10. At each 10-minute interval, one ribbon was taken out of the oven at a time and left to cool off naturally at ambient environment.

The experimental setup in which each magnetoelastic ribbon is tested for its magnetoelastic performance is shown in Figure 1b. A DC power supply is connected to the inner Helmholtz coils and produces the DC bias field, while a waveform generator is connected to the outer Helmholtz coils and produces the AC excitation field. The detection coil, located in the center of the Helmholtz coils, houses the magnetoelastic ribbon sensor and detects the sensor signal using the AC field. A pre-amplifier is then used to boost the signal, and its amplitude is measured with a KEITHLEY voltmeter.

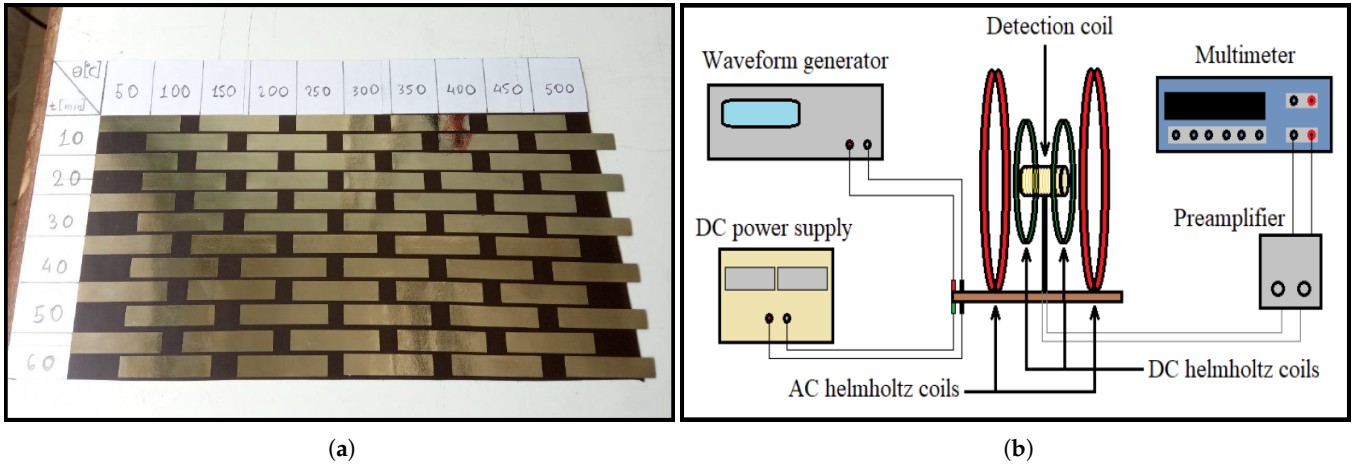

|         |         |
|:-------:|:-------:|
| (**a**) | (**b**) |

**Figure 1.** (**a**) The ribbons used for the annealing experiment and (**b**) the schematic of the experimental setup.

The setup works as follows: First, the ribbon is inserted inside the detection coil. The AC coil produces a small AC magnetic field $\Delta H = H_0\cos(2\pi ft)$, which is forcing the ribbon to vibrate at a fixed frequency of f = 1000 Hz (the choice of this frequency is justified below), while the DC bias coil produces a large fixed field $H_{DC}$, which brings the ribbon to a suitable operational biasing point (this is a standard technique in magnetoelastic-sensing applications as the biasing point gives a linear input0-output response and a better signal). The magnetoelastic nature of the ribbon causes it to develop a time-varying magnetization $\Delta M = M_0\cos(2\pi ft)$ (assuming a linear response for small signal analysis around the biasing point), which in turn creates a time-varying magnetic flux around the ribbon. A detection coil can sense this flux by means of Faraday's law of induction, resulting in an AC voltage $V = V_0\sin(2\pi ft)$ across it, which is amplified and measured by a voltmeter. This voltage is proportional to the time derivative of $\Delta M$ (this is the reason why we wrote V with a 90° out of phase to $\Delta M$), and from the definition of the susceptibility $\chi = \Delta M/\Delta H$, we can see that the magnitude $V_0$ is proportional to $\chi f H_0$. As f and $H_0$ are kept constant, an enhancement of the measured signal is expected when $\chi$ is enhanced, and vice-versa. The process of thermal annealing relaxes the internal stresses in the material, which are responsible for the high values of the anisotropy energy K [37] and subsequently for the high values of the anisotropy field $H_K$. The lowering of this field results in a higher susceptibility. Thus, this technique compares the voltage of each ribbon before and after annealing in order to find the optimum annealing parameters.

Shown in Figure 2 is the experimental setup for the magnetic characterization experiment, which basically produces the sample's hysteresis curve M–H where M is the sample magnetization and H the excitation field. The circuit consists of a resonance circuit, which is composed of a frequency generator together with set of capacitors and a large excitation coil, which together produce H; two detection coils, one empty and one containing the sample; an oscilloscope, and a PC where all data are collected and analyzed. The resonance circuit drives the excitation coil, which produces a large alternating magnetic field $H = H_0\cos(2\pi ft)$. The detection coil 2 (empty coil) produces a voltage $V_2$, which, according to Faraday's law, is proportional to the time derivative of H. Similarly, the detection coil 1 (the one containing the sample) produces a voltage $V_1$, which is proportional to the time derivative of the sample magnetization M (subtracting first a small term, which is proportional to H). These two voltages are collected by the oscilloscope, and the data are integrated in the PC to produce the desired M–H curve. The reason why the 1000 kHz frequency was chosen for the sensing experiments is the following: according to a previous work of our group [11], the magnetoelastic ribbons show an excellent flat frequency-response behavior in the range of 300 to 50,000 Hz, and so the frequency of 1000 kHz was a reasonable choice at the lower edge of this range.

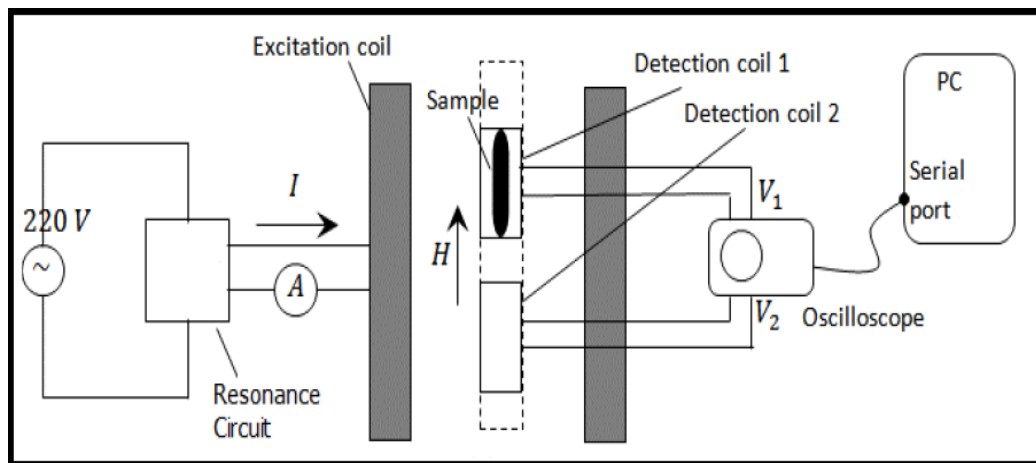

**Figure 2.** Schematic of the experimental setup of the magnetic characterization experiment.

### 3. Results and Discussion

#### 3.1. Sensing Properties

The sensing process was as follows: With no ribbon inside the detection coil of Figure 2, the amplifier was set to produce a fixed reading of 45.5 mV in the voltmeter, which represented the reference point. Subsequently, a non-annealed ribbon was inserted into the coil, and the new voltage $V_b = 65$ mV was recorded. After the annealing process, the annealed ribbons were inserted in the detection coil, and the quantity $\Delta V = V_a - V_b$ was recorded, where $V_a$ was the new voltage after the annealing process. This process was performed under two different conditions:

1. Without a bias field ($H_{DC} = 0$) during the sensing experiment.
2. With an optimum bias field $H_{DC}$ during the sensing experiment for signal enhancement.

##### 3.1.1. Absence of a Bias Field during the Sensing Experiments

Shown in Figure 3 is a 2D color map with the vertical axis being the annealing temperature, the horizontal axis the annealing time, and the color index representing the detected voltage $\Delta V$ without $H_{DC}$. The red color corresponds to high voltage values and the blue color to low voltage values. It is obvious that there is an optimum zone slightly above the material Curie temperature of $T_C = 350$ °C (for Metglas 2826 MB3) and below 450 °C where we obtained the better voltage signal. This zone is almost horizontal except at low times below 20 min where higher temperatures are necessary for the same results.

Note the color scale at the right where voltage differences as high as 19 mV are attainable. In comparison to the voltage produced by the non-annealed ribbon (V = 20.5 mV), we can obtain a maximum of $V_{max}$ = 39.5 mV, which is nearly a 100% improvement in the signal. Notice also that the zone after the time of 20 min is roughly centered around the crystallization temperature $T_{CR}$ = 410 °C, with the best results achieved either a little above it or close to it.

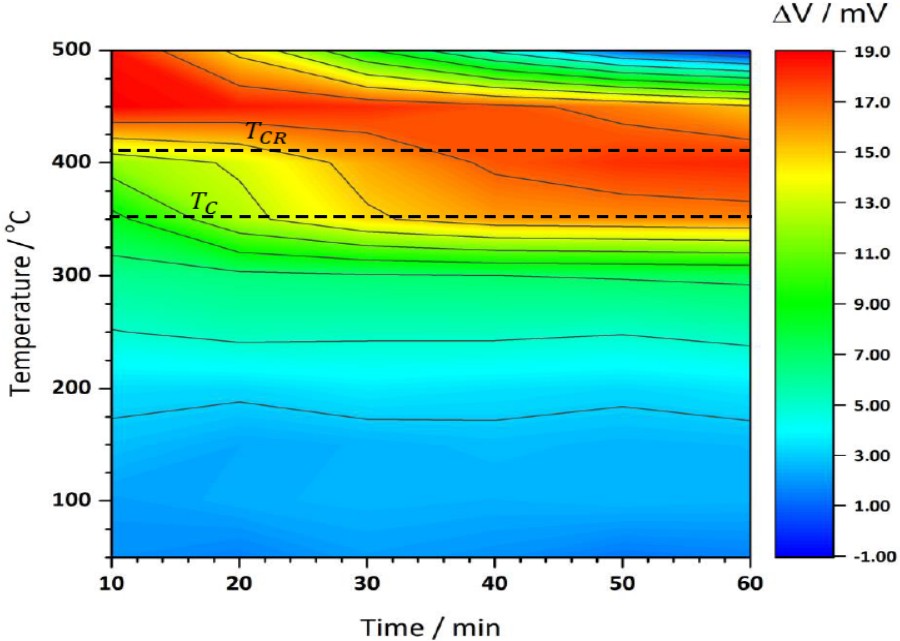

**Figure 3.** Map of annealing results for the case where there is no bias field ($H_{DC}$ = 0) during the sensing measurements. The color index represents the detected voltage ΔV. Shown with dashed lines is the Curie temperature $T_C$ = 353 °C and the crystallization temperature $T_{CR}$ = 410 °C.

3.1.2. Presence of a Bias Field during the Sensing Experiments

Shown in Figure 4 is the same 2D map as with Figure 3, but now the measurements of the voltage increase ΔV were done with the presence of a bias field $H_{DC}$. The same ribbons as with case 1 were used, but now the field $H_{DC}$ was adjusted in the range of 0.26–0.27 Gauss for maximum signal. Obviously, there are fundamental differences between the two cases. First of all, instead of a continuous zone of maximum signal (red color), there are now three distinct areas: one close to $T_C$ and very low times, one at 200 °C and relatively low times, and one at 200 °C and high times. Secondly, as the color scale shows, the maximum voltage difference is much less of the order of 3 mV, and that is a 15% improvement in the signal, which can be compared to the 100% improvement of the signal in the case 1. These results imply that we can achieve better sensing results even with no biasing ($H_{DC}$ = 0), provided that we make use of a properly annealed ribbon (for example, looking at Figure 3 we can have 400 °C for 1 h) compared to cases with $H_{DC} \neq 0$ and lower annealing temperatures (for example, 200 °C for 1 h at Figure 4). This is of great technological importance as we can have a simpler sensing apparatus with the elimination of the DC bias coil. For example, the magnetoelastic anti-theft tags always come in a pair with a little magnet so as to have them permanently biased for a better signal. This magnet can be eliminated with a properly annealed tag, which can lower the cost of the tag.

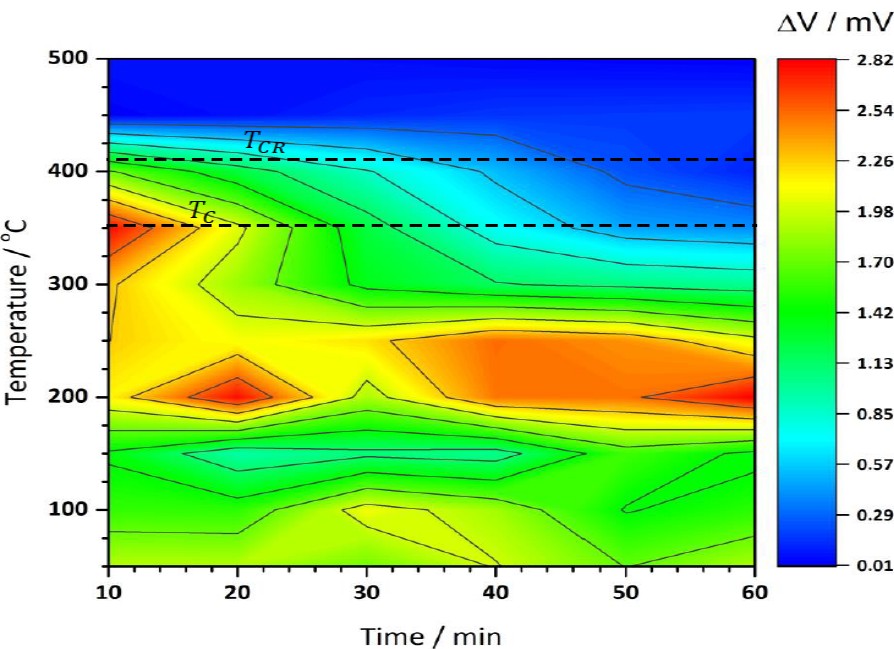

**Figure 4.** Map of annealing results for the case where a bias field ($H_{DC} \neq 0$) is applied during the sensing measurement. The color index represents the detected voltage $\Delta V$. Shown with dashed lines is the Curie temperature $T_C$ = 353 °C and the crystallization temperature $T_{CR}$ = 410 °C.

### 3.2. Magnetic Properties

Shown in Figure 5 are the hysteresis curves of six selected annealed ribbons, which were taken with AC fields at the frequency of 1000 Hz. From these curves, we can extract two important magnetic parameters, which are shown in the inset, and these are the coercivity $H_C$, which is the value of the applied magnetic field where the curve meets the horizontal axis, and the saturation magnetization $M_S$, which is the maximum magnetization. The value of $H_C$ is a measure of the squareness of the hysteresis loop and the magnetic hardness of the material, i.e., the higher the $H_C$, the harder the material is. The value of $M_S$, on the other hand, tells us how magnetic a sample is, i.e., the higher the value of $M_S$ is, the stronger the interaction of the material is with an existing magnetic field. The corresponding extracted data are shown in Table 1 together with the annealed times t and the temperatures $\theta$. The third column in the table shows the product of these two parameters, which we interpret as being an approximate measure of the heating energy that the samples received from the furnace. This is because the furnace temperature is proportional to its heating power, and this multiplied by time results in energy. Additionally, the last column shows the product $H_C \times M_S$ which is roughly equal to the area of the hysteresis loop, and it is numerically equal to the stored magnetic energy in the material.

Shown in Figure 6 is the $H_C$ and $M_S$ data versus the heat energy term t × $\theta$. It is obvious that as the offered heat increases, the material becomes softer (lowering of $H_C$) but magnetically stronger (rise of $M_S$). It is interesting to see that the values of the last column of Table 1, which, as was mentioned, represent the stored magnetic energy in the sample and remain quite constant except the last value. It is not clear why this value is so different from the other values, but if we assume that this is a discrepancy, then we can safely say that the stored energy in the sample is conserved during the annealing process. In other words, during the annealing process, the material reorganizes itself, decreasing anisotropy energy while increasing magnetostatic energy by better aligning local magnetic moments.

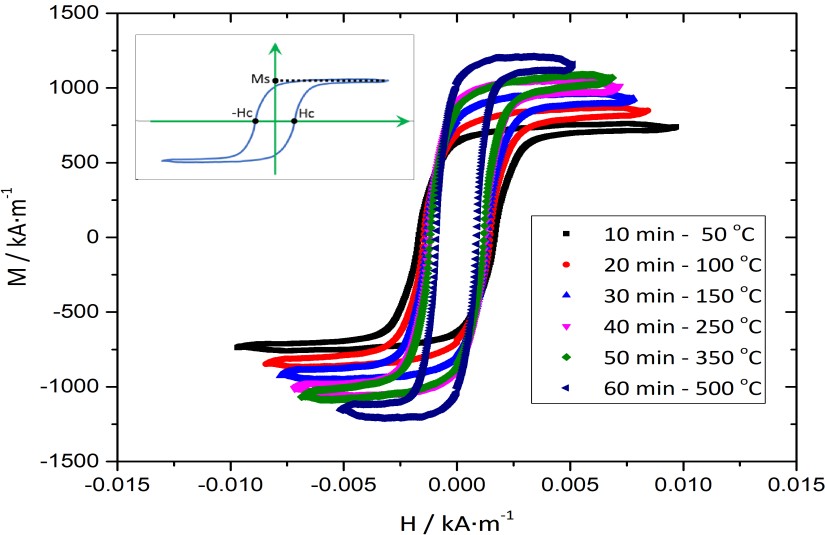

**Figure 5.** AC hysteresis curves for a set of the samples.

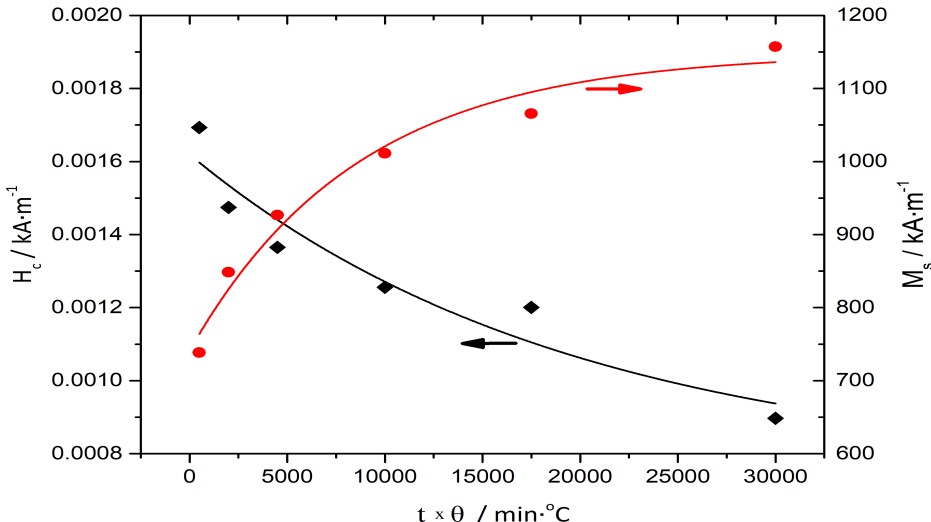

**Figure 6.** Coercivity $H_C$ and saturation magnetization $M_S$ data extracted from the hysteresis curves of Figure 5 versus the product $t \times \theta$.

**Table 1.** Coercivity data $H_C$ and saturation magnetization data $M_S$ extracted from Figure 5 graphs for a variety of samples with annealed times and temperatures.

| t (min) | $\theta$ (°C) | t × $\theta$ (min °C) | $H_C$ (kA/m) | $M_S$ (kA/m) | $H_C \times M_S$ (kA/m)$^2$ |
|---|---|---|---|---|---|
| 10 | 50 | 500 | 0.00169 | 738 | 1.25 |
| 20 | 100 | 2000 | 0.00147 | 849 | 1.25 |
| 30 | 150 | 4500 | 0.00137 | 927 | 1.26 |
| 40 | 200 | 10,000 | 0.00126 | 1010 | 1.27 |
| 50 | 250 | 17,500 | 0.00120 | 1070 | 1.28 |
| 60 | 300 | 30,000 | 0.00090 | 1160 | 1.04 |

*3.3. Comparison to Previous Works*

Shown in Table 2 is a summary of the results of the references [31–36] presented in the introduction section together with our results (last line). The second column shows the Metglas material studied in these works; the third column shows the Curie temperature T (°C) of these materials; the fourth column shows the optimum annealing temperature; the fifth and sixth columns show the change in the $M_S$ and $H_C$ parameters, respectively;

and the last column shows the change in other parameters. In our case, we obtained the ratios of the last to first points of Figure 6 for the $M_S$ and $H_C$ changes. It can be seen that our values are comparable to those given by other authors; for example, Ref. [36] reports exactly the same $H_C$ change and Ref. [31] about the same $M_S$ change. Except for one case, annealing improves the magnetic properties, resulting in smaller hysteresis loops, higher magnetization values, lower losses, and less anisotropy.

**Table 2.** Comparison results.

| Ref. | Metglas | $T_C$ (°C) | $T_A$ (°C) | $M_S$ Ratio | $H_C$ Ratio | Other |
|---|---|---|---|---|---|---|
| [31] | $Fe_{78}B_{13}Si_9$ | 415 | 345 | 1.8 | 0.2 | - |
| [32] | $Fe_{40}Ni_{40}B_{14}B_6$ | - | 350 | - | - | Power loss reduction |
| [33] | $Fe_{64}Ni_{10}Nb_3CuSi_{13}B_9$ | - | 520 | Const. | 0.1 | Lower anisotropy |
| [34] | $Fe_{45}Ni_{45}Mo_7B_3$ | 353 | 250 | - | - | Quality factor increase |
| [35] | $Fe_{78}B_9Si_{13}$ | 508 | 450 | - | - | No improvement |
| [36] | $Fe_{78}B_{13}Si_9$ | 415 | 500 | - | 0.5 | - |
| This work | $Fe_{40}Ni_{38}Mo_4B_{18}$ | 353 | 400 | 1.6 | 0.5 | Output signal increase |

## 4. Conclusions

The process of the thermal annealing is definitely enhancing the sensing ability of the magnetoelastic materials, so it is a helpful process for sensor technology. Our results for the smart material Metglas 2826 MB3 show that a 100% improvement is possible to the sensing signal without a biasing field, by annealing the samples at a temperature zone that lies between the Curie temperature of 350 °C and the crystallization temperature of $T_{CR}$ = 410 °C, for a time interval of 30 to 60 min. Applying a biasing field during the sensing experiment brings a smaller signal improvement of 15%, but it lowers significantly the annealing temperature to 200 °C and the annealing time to 20 min. The magnetic measurements show a decrease in the $H_C$ value by about 2 and a corresponding increase of $M_S$ by about 1.6, which is consistent with the work of other groups. Additionally, it appears that the stored energy in the sample is conserved during the annealing process, possibly because the material reorganizes itself and lowers its anisotropy energy but at the same time increases its magnetostatic energy. We believe that our work is of great technological importance as with proper annealing, the DC bias field can be eliminated, thus using a simpler sensing apparatus.

**Author Contributions:** Conceptualization, G.S., K.V., and D.K.; methodology, G.S. and K.V.; software, G.S.; validation G.S. and D.K.; formal analysis, G.S. and D.K.; investigation, G.S. and K.V.; resources, D.K.; data curation, G.S.; writing—original draft preparation, D.K.; writing—review and editing, G.S. and D.K.; visualization, G.S.; supervision, D.K.; project administration, D.K.; funding acquisition, D.K. All authors have read and agreed to the published version of the manuscript.

**Funding:** This research received no external funding.

**Institutional Review Board Statement:** Not applicable.

**Informed Consent Statement:** Not applicable.

**Data Availability Statement:** Not applicable.

**Conflicts of Interest:** The authors declare no conflict of interest.

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
