# Peer review of "The Effect of the Thermal Annealing Process to the Sensing Performance of Magnetoelastic Ribbon Materials"

_sustainability, doi:10.3390/su132413947_

Round 1
Reviewer 1 Report
The authors prepared the manuscript entitled as “The effect of the thermal annealing process to the sensing performance of magnetoelastic ribbon materials” which has some useful informations about the experimental analysis and conclusions. However, the paper has no novelty, and requires many improvements about the presentation and main text.
- The novelty of the paper is missing in the abstract. What is the importance of the paper? What is the practical significance? Please show these first in the abstract, also add them to the last paragraph of the introduction and conclusion.
- It must be noted that language of the paper is weak, hard to understand need to be written from the start to the end.
- Referencing of the paper is a problem. Given references such as 2-5, 8-12, cannot explain these studies. What did they find in their results? These results have to be addressed.
- No need for actual and schematic photos together. Experimental setup must be given in detail.
- There is no in depth analysis of the results. Also, obtained findings have to be analyzed with the support of the current literature works.
- Conclusion section is weak. The key findings have to be added with more detail. Also, the mechanisms during obtaining of the results must be included into the conclusions.
- There is a problem in the references. If the subject of the paper fits for the journals indexing area, then the authors have to add many papers published in Sustainability.
Author Response
Please see the attachment for highlighted changes
We would like to thank the referee for providing very useful comments as they help us to improve our manuscript. This pdf file has all of the responses to referee comments, while the manuscript is included in the second pdf file, along with all of the changes, highlighted and commented.
Comments Issue 1: The novelty of the paper is missing in the abstract. What is the importance of the paper? What is the practical significance? Please show these first in the abstract, also add them to the last paragraph of the introduction and conclusion.
- Authors answer: The abstract and conclusion sections have been reconstructed in order to reflect better the novelty, the importance and the practical significance of the paper.
Issue 2: It must be noted that language of the paper is weak, hard to understand need to be written from the start to the end.
- Authors answer: The entire manuscript has been revised for English grammar and syntax.
Issue 3: Referencing of the paper is a problem. Given references such as 2-5, 8-12, cannot explain these studies. What did they find in their results? These results have to be addressed.
- Authors answer: With all due respect, we are unsure what the referee means by this comment. These references are provided not to explain our research, but to emphasize the importance of magnetoelastic sensors in a variety of applications such as damage detection, biosensing, biomedical engineering, food processing, and medicine, among others. We have redrawn the lines to emphasize this point.
Issue 4: No need for actual and schematic photos together. Experimental setup must be given in detail.
- Authors answer: Actual photos have been removed. In addition, a paragraph was added to the experimental section to better describe the experimental setup.
Issue 5: There is no in-depth analysis of the results. Also, obtained findings have to be analyzed with the support of the current literature works.
- Authors answer: Please see answer to next comment.
Issue 6: Conclusion section is weak. The key findings have to be added with more detail. Also, the mechanisms during obtaining of the results must be included into the conclusions.
- Authors answer: In the Results and discussion section, a new subsection was added that includes an in-depth analysis of key findings as well as comparisons with other works of literature. In addition, a few lines have been added to the Conclusion section.
Issue 7: There is a problem in the references. If the subject of the paper fits for the journals indexing area, then the authors have to add many papers published in Sustainability.
- Authors answer: Articles published in Sustainability were included to the first paragraph of the introduction section.

Reviewer 2 Report
I suggest same corrections.
Page 1: 1 Introduction: Line 24: The magnetic response is also very important in the highly sensitive quartz method for detecting very weak magnetic changes in the magnetic field, as shown in ref .:
-Matko, V., Milanovič, M. High resolution switching mode inductance-to-frequency converter with temperature compensation. Sensors, ISSN 1424-8220, 2014, 14, 10, 19242-19259. https://www.mdpi.com/1424-8220/14/10/19242
-Matko V., Šafarič R. Major improvements of quartz crystal pulling sensitivity and linearity using series reactance. Sensors, 2009, 9, 10, 8263-8270. https://www.ncbi.nlm.nih.gov/pmc/articles/PMC3292106/
Athors should include the references above in the paper.
Page 4: Figure 2a: This picture is supposed to be bigger.
Figure 2b: What are the values of the inductances of the detection coils?
Page 4: By how much the sensing properties are improved by the proposed procedure in the article
Page 7: Are the hysteresis curves temperature-dependent after changing the ambient temperature after the completion of the anneling process?
Author Response
Please see the attachment for highlighted changes
We would like to thank the referee for providing very useful comments as they help us to improve our manuscript. This pdf file has all of the responses to referee comments, while the manuscript is included in the second pdf file, along with all of the changes, highlighted and commented.
Comments Issue 1: Authors should include the references above in the paper.
- Authors answer: The suggested articles were added to the first paragraph of the introduction.
Issue 2: Figure 2a should be bigger.
- Authors answer: Figure 2a removed and figure 2b resized.
Issue 3: In figure 2b what are the values of the inductances of the detection coils?
- Authors answer: Both detection coils have the same value of inductance, which is L=3 mH.
Issue 4: By how much the sensing properties are improved by the proposed procedure in the article.
- Authors answer: The sensor signal can be improved by up to 100%, as noted in the abstract and conclusion section.
Issue 5: Are the hysteresis curves temperature-dependent after changing the ambient temperature after the completion of the annealing process.
- Authors answer: Upon completion of the annealing process, no such changes were observed

Reviewer 3 Report
This manuscript presents information on the investigation of optimal annealing parameters for a class of intelligent magnetoelastic and sensor materials, in particular the Metglas 2826 MB alloy. The work contains experimental discussions of the sensing and magnetic properties of annealed and unannealed ribbons of the alloy under study with and without an optimal bias field. The results shown in the manuscript can be applied to improve load and vibration sensor technologies as well as probes for medicine and gas research due to their magnetic response to mechanical strain. The article is clearly written and the conclusions are adequately supported by the data presented.
However, there are a few points that need to be corrected before publication of this manuscript.
The following minor issue needs to be addressed:
- In Chapter 3. Results and discussion you use the phrase "the more magnetic the material is." Please define which material is "more magnetic. Clarify in the text what the author meant to say?
- In References, please add more contemporary publications on your subject.
Major comments:
- Could you please provide hysteresis loops for magnetosaturated samples: in Fig. 5 the hysteresis loops are not fully saturated and it is not possible to extract real Ms values from this data.
Author Response
Please see the attachment for highlighted changes
We would like to thank the referee for providing very useful comments as they help us to improve our manuscript. This pdf file has all of the responses to referee comments, while the manuscript is included in the second pdf file, along with all of the changes, highlighted and commented.
Comments Issue 1: Results and discussion you use the phrase "the more magnetic the material is." Please define which material is "more magnetic. Clarify in the text what the author meant to say?
- Authors answer: Line was rephrased in response to reviewer suggestion.
Issue 2: In References, please add more contemporary publications on your subject.
- Authors answer: Some contemporary articles were added to the reference section.
Issue 3: Could you please provide hysteresis loops for magneto saturated samples: in Fig. 5 the hysteresis loops are not fully saturated and it is not possible to extract real Ms values from this data.
- Authors answer: The driving field was oscillating in the range of [-0.010 +0.010] kA/m for all hysteresis loops. All loops were saturated within this range, thus we plotted the curves until there was an overlap near the saturation points for better visualization between them.

Round 2
Reviewer 1 Report
The authors prepared the paper named “The effect of the thermal annealing process to the sensing performance of magnetoelastic ribbon materials” in reviewed version. I have reviewed the paper and see that the authors have been made many contributions to article. I think the paper is ready for publishing at the final version. My decision is about to accept it.
